# IMPROVING ZERO-SHOT VOICE STYLE TRANSFER VIA DISENTANGLED REPRESENTATION LEARNING

**Siyang Yuan**[1*]**, Pengyu Cheng**[1*]**, Ruiyi Zhang**[1]**, Weituo Hao**[1]**, Zhe Gan**[2] **and Lawrence Carin**[1]
[1]Duke University, Durham, North Carolina, USA
[2]Microsoft, Redmond, Washington, USA
{`siyang.yuan,pengyu.cheng`}@duke.edu

## ABSTRACT

Voice style transfer, also called voice conversion, seeks to modify one speaker's voice to generate speech as if it came from another (target) speaker. Previous works have made progress on voice conversion with parallel training data and pre-known speakers. However, zero-shot voice style transfer, which learns from non-parallel data and generates voices for previously unseen speakers, remains a challenging problem. We propose a novel zero-shot voice transfer method via disentangled representation learning. The proposed method first encodes speaker-related *style* and voice *content* of each input voice into separated low-dimensional embedding spaces, and then transfers to a new voice by combining the source content embedding and target style embedding through a decoder. With information-theoretic guidance, the style and content embedding spaces are representative and (ideally) independent of each other. On real-world VCTK datasets, our method outperforms other baselines and obtains state-of-the-art results in terms of transfer accuracy and voice naturalness for voice style transfer experiments under both many-to-many and zero-shot setups.

## 1 INTRODUCTION

Style transfer, which automatically converts a data instance into a target style, while preserving its content information, has attracted considerable attention in various machine learning domains, including computer vision (Gatys et al., 2016; Luan et al., 2017; Huang & Belongie, 2017), video processing (Huang et al., 2017; Chen et al., 2017), and natural language processing (Shen et al., 2017; Yang et al., 2018; Lample et al., 2019; Cheng et al., 2020b). In speech processing, style transfer was earlier recognized as voice conversion (VC) (Muda et al., 2010), which converts one speaker's utterance, as if it was from another speaker but with the same semantic meaning. Voice style transfer (VST) has received long-term research interest, due to its potential for applications in security (Sisman et al., 2018), medicine (Nakamura et al., 2006), entertainment (Villavicencio & Bonada, 2010) and education (Mohammadi & Kain, 2017), among others.

Although widely investigated, VST remains challenging when applied to more general application scenarios. Most of the traditional VST methods require parallel training data, *i.e.*, paired voices from two speakers uttering the same sentence. This constraint limits the application of such models in the real world, where data are often not pair-wise available. Among the few existing models that address non-parallel data (Hsu et al., 2016; Lee & Wu, 2006; Godoy et al., 2011), most methods cannot handle many-to-many transfer (Saito et al., 2018; Kaneko & Kameoka, 2018; Kameoka et al., 2018), which prevents them from converting multiple source voices to multiple target speaker styles. Even among the few non-parallel many-to-many transfer models, to the best of our knowledge, only two models (Qian et al., 2019; Chou & Lee, 2019) allow zero-shot transfer, *i.e.*, conversion from/to newly-coming speakers (unseen during training) without re-training the model.

The only two zero-shot VST models (AUTOVC (Qian et al., 2019) and AdaIN-VC (Chou & Lee, 2019)) share a common weakness. Both methods construct encoder-decoder frameworks, which extract the style and the content information into style and content embeddings, and generate a voice sample by combining a style embedding and a content embedding through the decoder. With the combination of the source content embedding and the target style embedding, the models generate

---
*Equal contribution.

the transferred voice, based only on source and target voice samples. AUTOVC (Qian et al., 2019) uses a GE2E (Wan et al., 2018) pre-trained style encoder to ensure rich speaker-related information in style embeddings. However, AUTOVC has no regularizer to guarantee that the content encoder does not encode any style information. AdaIN-VC (Chou & Lee, 2019) applies instance normalization (Ulyanov et al., 2016) to the feature map of content representations, which helps to eliminate the style information from content embeddings. However, AdaIN-VC fails to prevent content information from being revealed in the style embeddings. Both methods cannot assure that the style and content embeddings are disentangled without information revealed from each other.

With information-theoretic guidance, we propose a disentangled-representation-learning method to enhance the encoder-decoder zero-shot VST framework, for both style and content information preservation. We call the proposed method **I**nformation-theoretic **D**isentangled **E**mbedding for **V**oice **C**onversion (IDE-VC). Our model successfully induces the style and content of voices into independent representation spaces by minimizing the mutual information between style and content embeddings. We also derive two new multi-group mutual information lower bounds, to further improve the representativeness of the latent embeddings. Experiments demonstrate that our method outperforms previous works under both many-to-many and zero-shot transfer setups on two objective metrics and two subjective metrics.

## 2 BACKGROUND

In information theory, mutual information (MI) is a crucial concept that measures the dependence between two random variables. Mathematically, the MI between two variables $\boldsymbol{x}$ and $\boldsymbol{y}$ is

$$\mathcal{I}(\boldsymbol{x}; \boldsymbol{y}) := \mathbb{E}_{p(\boldsymbol{x},\boldsymbol{y})}\Big[\log \frac{p(\boldsymbol{x}, \boldsymbol{y})}{p(\boldsymbol{x})p(\boldsymbol{y})}\Big], \tag{1}$$

where $p(\boldsymbol{x})$ and $p(\boldsymbol{y})$ are marginal distributions of $\boldsymbol{x}$ and $\boldsymbol{y}$, and $p(\boldsymbol{x}, \boldsymbol{y})$ is the joint distribution. Recently, MI has attracted considerable interest in machine learning as a criterion to minimize or maximize the dependence between different parts of a model (Chen et al., 2016; Alemi et al., 2016; Hjelm et al., 2018; Veličković et al., 2018; Song et al., 2019). However, the calculation of exact MI values is challenging in practice, since the closed form of joint distribution $p(\boldsymbol{x}, \boldsymbol{y})$ in equation (1) is generally unknown. To solve this problem, several MI estimators have been proposed. For MI maximization tasks, Nguyen, Wainwright and Jordan (NWJ) (Nguyen et al., 2010) propose a lower bound by representing (1) as an $f$-divergence (Moon & Hero, 2014):

$$\mathcal{I}_{\text{NWJ}} := \mathbb{E}_{p(\boldsymbol{x},\boldsymbol{y})}[f(\boldsymbol{x}, \boldsymbol{y})] - e^{-1}\mathbb{E}_{p(\boldsymbol{x})p(\boldsymbol{y})}[e^{f(\boldsymbol{x},\boldsymbol{y})}], \tag{2}$$

with a score function $f(\boldsymbol{x}, \boldsymbol{y})$. Another widely-used sample-based MI lower bound is InfoNCE (Oord et al., 2018), which is derived with Noise Contrastive Estimation (NCE) (Gutmann & Hyvärinen, 2010). With sample pairs $\{(\boldsymbol{x}_i, \boldsymbol{y}_i)\}_{i=1}^{N}$ drawn from the joint distribution $p(\boldsymbol{x}, \boldsymbol{y})$, the InfoNCE lower bound is defined as

$$\mathcal{I}_{\text{NCE}} := \mathbb{E}\Big[\frac{1}{N}\sum_{i=1}^{N}\log\frac{e^{f(\boldsymbol{x}_i,\boldsymbol{y}_i)}}{\frac{1}{N}\sum_{j=1}^{N}e^{f(\boldsymbol{x}_i,\boldsymbol{y}_j)}}\Big]. \tag{3}$$

For MI minimization tasks, Cheng et al. (2020a) proposed a contrastively learned upper bound that requires the conditional distribution $p(\boldsymbol{x}|\boldsymbol{y})$:

$$\mathcal{I}(\boldsymbol{x}; \boldsymbol{y}) \leq \mathbb{E}\Big[\frac{1}{N}\sum_{i=1}^{N}\Big[\log p(\boldsymbol{x}_i|\boldsymbol{y}_i) - \frac{1}{N}\sum_{j=1}^{N}\log p(\boldsymbol{x}_j|\boldsymbol{y}_i)\Big]\Big]. \tag{4}$$

where the MI is bounded by the log-ratio of conditional distribution $p(\boldsymbol{x}|\boldsymbol{y})$ between positive and negative sample pairs. In the following, we derive our information-theoretic disentangled representation learning framework for voice style transfer based on the MI estimators described above.

## 3 PROPOSED MODEL

We assume access to $N$ audio (voice) recordings from $M$ speakers, where speaker $u$ has $N_u$ voice samples $\mathcal{X}_u = \{\boldsymbol{x}_{ui}\}_{i=1}^{N_u}$. The proposed approach encodes each voice input $\boldsymbol{x} \in \mathcal{X} = \cup_{u=1}^{M}\mathcal{X}_u$ into a speaker-related (style) embedding $\boldsymbol{s} = E_s(\boldsymbol{x})$ and a content-related embedding $\boldsymbol{c} = E_c(\boldsymbol{x})$,

using respectively a style encoder $E_s(\cdot)$ and a content encoder $E_c(\cdot)$. To transfer a source $\boldsymbol{x}_{ui}$ from speaker $u$ to the target style of the voice of speaker $v$, $\boldsymbol{x}_{vj}$, we combine the content embedding $\boldsymbol{c}_{ui} = E_c(\boldsymbol{x}_{ui})$ and the style embedding $\boldsymbol{s}_{vj} = E_s(\boldsymbol{x}_{vj})$ to generate the transferred voice $\hat{\boldsymbol{x}}_{u \to v,i} = D(\boldsymbol{s}_{vj}, \boldsymbol{c}_{ui})$ with a decoder $D(\boldsymbol{s}, \boldsymbol{c})$. To implement this two-step transfer process, we introduce a novel mutual information (MI)-based learning objective, that induces the style embedding $\boldsymbol{s}$ and content embedding $\boldsymbol{c}$ into independent representation spaces (*i.e.*, ideally, $\boldsymbol{s}$ contains rich style information of $\boldsymbol{x}$ with no content information, and *vice versa*). In the following, we first describe our MI-based training objective in Section 3.1, and then discuss the practical estimation of the objective in Sections 3.2 and 3.3.

## 3.1 MI-BASED DISENTANGLING OBJECTIVE

From an information-theoretic perspective, to learn representative latent embedding $(\boldsymbol{s}, \boldsymbol{c})$, it is desirable to maximize the mutual information between the embedding pair $(\boldsymbol{s}, \boldsymbol{c})$ and the input $\boldsymbol{x}$. Meanwhile, the style embedding $\boldsymbol{s}$ and the content $\boldsymbol{c}$ are desired to be independent, so that we can control the style transfer process with different style and content attributes. Therefore, we minimize the mutual information $\mathcal{I}(\boldsymbol{s}; \boldsymbol{c})$ to disentangle the style embedding and content embedding spaces. Consequently, our overall disentangled-representation-learning objective seeks to minimize

$$\mathcal{L} = \mathcal{I}(\boldsymbol{s}; \boldsymbol{c}) - \mathcal{I}(\boldsymbol{x}; \boldsymbol{s}, \boldsymbol{c}) = \mathcal{I}(\boldsymbol{s}; \boldsymbol{c}) - \mathcal{I}(\boldsymbol{x}; \boldsymbol{c}|\boldsymbol{s}) - \mathcal{I}(\boldsymbol{x}; \boldsymbol{s}). \tag{5}$$

As discussed in Locatello *et al.* (Locatello et al., 2019), without inductive bias for supervision, the learned representation can be meaningless. To address this problem, we use the speaker identity $\boldsymbol{u}$ as a variable with values $\{1, \dots, M\}$ to learn representative style embedding $\boldsymbol{s}$ for speaker-related attributes. Noting that the process from speaker $u$ to his/her voice $\boldsymbol{x}_{ui}$ to the style embedding $\boldsymbol{s}_{ui}$ (as $\boldsymbol{u} \to \boldsymbol{x} \to \boldsymbol{s}$) is a Markov Chain, we conclude $\mathcal{I}(\boldsymbol{s}; \boldsymbol{x}) \geq \mathcal{I}(\boldsymbol{s}; \boldsymbol{u})$ based on the MI data-processing inequality (Cover & Thomas, 2012) (as stated in the Supplementary Material). Therefore, we replace $\mathcal{I}(\boldsymbol{s}; \boldsymbol{x})$ in $\mathcal{L}$ with $\mathcal{I}(\boldsymbol{s}; \boldsymbol{u})$ and minimize an upper bound instead:

$$\bar{\mathcal{L}} = \mathcal{I}(\boldsymbol{s}; \boldsymbol{c}) - \mathcal{I}(\boldsymbol{x}; \boldsymbol{c}|\boldsymbol{s}) - \mathcal{I}(\boldsymbol{u}; \boldsymbol{s}) \geq \mathcal{I}(\boldsymbol{s}; \boldsymbol{c}) - \mathcal{I}(\boldsymbol{x}; \boldsymbol{c}|\boldsymbol{s}) - \mathcal{I}(\boldsymbol{x}; \boldsymbol{s}), \tag{6}$$

In practice, calculating the MI is challenging, as we typically only have access to samples, and lack the required distributions (Chen et al., 2016). To solve this problem, below we provide several MI estimates to the objective terms $\mathcal{I}(\boldsymbol{s}; \boldsymbol{c})$, $\mathcal{I}(\boldsymbol{x}; \boldsymbol{c}|\boldsymbol{s})$ and $\mathcal{I}(\boldsymbol{u}; \boldsymbol{s})$.

## 3.2 MI LOWER BOUND ESTIMATION

To maximize $\mathcal{I}(\boldsymbol{u}; \boldsymbol{s})$, we derive the following multi-group MI lower bound (Theorem 3.1) based on the NWJ bound developed in Nguyen *et al.* (Nguyen et al., 2010). The detailed proof is provided in the Supplementary Material. Let $\boldsymbol{\mu}_v^{(-ui)} = \boldsymbol{\mu}_v$ represent the mean of all style embeddings in group $\mathcal{X}_v$, constituting the style centroid of speaker $v$; $\boldsymbol{\mu}_u^{(-ui)}$ is the mean of all style embeddings in group $\mathcal{X}_u$ except data point $\boldsymbol{x}_{ui}$, representing a leave-$\boldsymbol{x}_{ui}$-out style centroid of speaker $u$. Intuitively, we minimize $\|\boldsymbol{s}_{ui} - \boldsymbol{\mu}_u^{(-ui)}\|$ to encourage the style embedding of voice $\boldsymbol{x}_{ui}$ to be more similar to the style centroid of speaker $u$, while maximizing $\|\boldsymbol{s}_{ui} - \boldsymbol{\mu}_v^{(-ui)}\|$ to enlarge the margin between $\boldsymbol{s}_{ui}$ and the other speakers' style centroids $\boldsymbol{\mu}_v$. We denote the right-hand side of (7) as $\hat{\mathcal{I}}_1$.

**Theorem 3.1.** *Let* $\boldsymbol{\mu}_v^{(-ui)} = \frac{1}{N_v} \sum_{k=1}^{N_v} \boldsymbol{s}_{vk}$ *if* $u \neq v$; *and* $\boldsymbol{\mu}_u^{(-ui)} = \frac{1}{N_u - 1} \sum_{j \neq i} \boldsymbol{s}_{uj}$. *Then,*

$$\mathcal{I}(\boldsymbol{u}; \boldsymbol{s}) \geq \mathbb{E}\Big[\frac{1}{N} \sum_{u=1}^{M} \sum_{i=1}^{N_u} \Big[ -\|\boldsymbol{s}_{ui} - \boldsymbol{\mu}_u^{(-ui)}\|^2 - \frac{e^{-1}}{N} \sum_{v=1}^{M} N_v \exp\{-\|\boldsymbol{s}_{ui} - \boldsymbol{\mu}_v^{(-ui)}\|^2\}\Big]\Big]. \tag{7}$$

To maximize $\mathcal{I}(\boldsymbol{x}; \boldsymbol{c}|\boldsymbol{s})$, we derive a conditional mutual information lower bound below:

**Theorem 3.2.** *Assume that given* $\boldsymbol{s} = \boldsymbol{s}_u$, *samples* $\{(\boldsymbol{x}_{ui}, \boldsymbol{c}_{ui})\}_{i=1}^{N_u}$ *are observed. With a variational distribution* $q_\phi(\boldsymbol{x}|\boldsymbol{s}, \boldsymbol{c})$, *we have* $\mathcal{I}(\boldsymbol{x}; \boldsymbol{c}|\boldsymbol{s}) \geq \mathbb{E}[\hat{\mathcal{I}}]$, *where*

$$\hat{\mathcal{I}} = \frac{1}{N} \sum_{u=1}^{M} \sum_{i=1}^{N_u} \Big[ \log q_\phi(\boldsymbol{x}_{ui}|\boldsymbol{c}_{ui}, \boldsymbol{s}_u) - \log\Big(\frac{1}{N_u} \sum_{j=1}^{N_u} q_\phi(\boldsymbol{x}_{uj}|\boldsymbol{c}_{ui}, \boldsymbol{s}_u)\Big)\Big]. \tag{8}$$

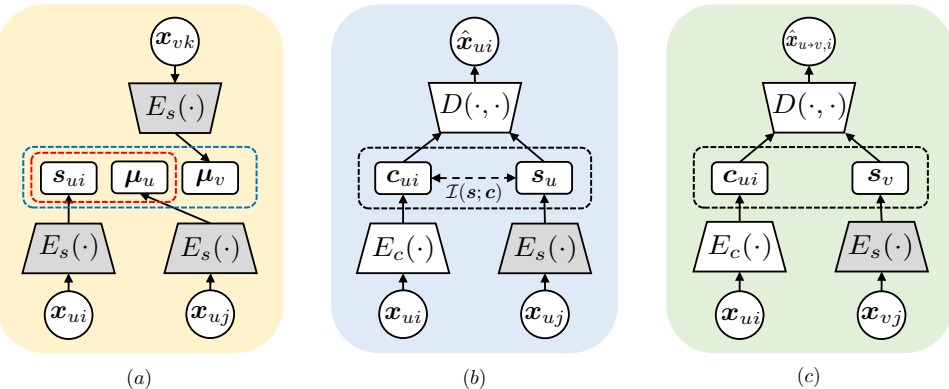

Figure 1: Training and transfer processes. (a) Training style encoder $E_s$ with objective $\hat{\mathcal{I}}_1$: All voice samples are encoded into style embedding space. For style embedding $\boldsymbol{s}_{ui}$ of $\boldsymbol{x}_{ui}$, we minimize its distance with speaker $u$'s style centroid $\boldsymbol{\mu}_u$, and maximize its distance to other speaker style centroids $\boldsymbol{\mu}_v$. (b) Training for content encoder $E_c$ and decoder $D$ as objectives $\hat{\mathcal{I}}_2, \hat{\mathcal{I}}_3$: We encode content $\boldsymbol{c}_{ui}$ from voice $\boldsymbol{x}_{ui}$ from speaker $u$. The style of speaker $u$ is encoded from another speaker $u$'s voice $\boldsymbol{x}_{uj}$. The dependency of style and content embedding is minimized with $\hat{\mathcal{I}}_3$. With $\boldsymbol{c}_{ui}$ and $\boldsymbol{s}_u$, the decoder reconstructs the voice $\boldsymbol{x}_{ui}$ as $\hat{\boldsymbol{x}}_{ui} = D(\boldsymbol{s}_u, \boldsymbol{c}_{ui})$. Then $\hat{\mathcal{I}}_2$ is calculated based on the original voice $\boldsymbol{c}_{ui}$ and the reconstruction $\hat{\boldsymbol{c}}_{ui}$. (c) Transfer process: for zero-shot voice style transfer, with $\boldsymbol{x}_{ui}$ from speaker $u$ and $\boldsymbol{x}_{vj}$ from speaker $v$, we encode content $\boldsymbol{c}_{ui}$ and style $\boldsymbol{s}_v$, and combine them together to generate a transferred voice $\hat{\boldsymbol{x}}_{u \to v, i} = D(\boldsymbol{s}_v, \boldsymbol{c}_{ui})$.

Based on the criterion for $\boldsymbol{s}$ in equation (7), a well-learned style encoder $E_s$ pulls all style embeddings $\boldsymbol{s}_{ui}$ from speaker $u$ together. Suppose $\boldsymbol{s}_u$ is representative of the style embeddings of set $\mathcal{X}_u$. If we parameterize the distribution $q_\phi(\boldsymbol{x}|\boldsymbol{s}, \boldsymbol{c}) \propto \exp(-\|\boldsymbol{x} - D(\boldsymbol{s}, \boldsymbol{c})\|^2)$ with decoder $D(\boldsymbol{s}, \boldsymbol{c})$, then based on Theorem 3.2, we can estimate the lower bound of $\mathcal{I}(\boldsymbol{x}; \boldsymbol{c}|\boldsymbol{s})$ with the following objective:

$$\hat{\mathcal{I}}_2 := \frac{1}{N} \sum_{u=1}^{M} \sum_{i=1}^{N_u} \Big[ -\|\boldsymbol{x}_{ui} - D(\boldsymbol{c}_{ui}, \boldsymbol{s}_u)\|^2 - \log\Big( \frac{1}{N_u} \sum_{j=1}^{N_u} \exp\{-\|\boldsymbol{x}_{uj} - D(\boldsymbol{c}_{ui}, \boldsymbol{s}_u)\|^2\} \Big) \Big].$$

When maximizing $\hat{\mathcal{I}}_2$, for speaker $u$ with his/her given voice style $\boldsymbol{s}_u$, we encourage the content embedding $\boldsymbol{c}_{ui}$ to well reconstruct the original voice $\boldsymbol{x}_{ui}$, with small $\|\boldsymbol{x}_{ui} - D(\boldsymbol{c}_{ui}, \boldsymbol{s}_u)\|$. Additionally, the distance $\|\boldsymbol{x}_{uj} - D(\boldsymbol{c}_{ui}, \boldsymbol{s}_u)\|$ is minimized, ensuring $\boldsymbol{c}_{ui}$ does not contain information to reconstruct other voices $\boldsymbol{x}_{uj}$ from speaker $u$. With $\hat{\mathcal{I}}_2$, the correlation between $\boldsymbol{x}_{ui}$ and $\boldsymbol{c}_{ui}$ is amplified, which improves $\boldsymbol{c}_{ui}$ in preserving the content information.

### 3.3 MI UPPER BOUND ESTIMATION

The crucial part of our framework is disentangling the style and the content embedding spaces, which imposes (ideally) that the style embedding $\boldsymbol{s}$ excludes any content information and *vice versa*. Therefore, the mutual information between $\boldsymbol{s}$ and $\boldsymbol{c}$ is expected to be minimized. To estimate $\mathcal{I}(\boldsymbol{s}; \boldsymbol{c})$, we derive a sample-based MI *upper* bound in Theorem 3.3 base on (4).

**Theorem 3.3.** *If $p(\boldsymbol{s}|\boldsymbol{c})$ provides the conditional distribution between variables $\boldsymbol{s}$ and $\boldsymbol{c}$, then*

$$\mathcal{I}(\boldsymbol{s}; \boldsymbol{c}) \le \mathbb{E}\Big[ \frac{1}{N} \sum_{u=1}^{M} \sum_{i=1}^{N_u} \Big[ \log p(\boldsymbol{s}_{ui}|\boldsymbol{c}_{ui}) - \frac{1}{N} \sum_{v=1}^{M} \sum_{j=1}^{N_v} \log p(\boldsymbol{s}_{ui}|\boldsymbol{c}_{vj}) \Big] \Big]. \tag{9}$$

The upper bound in (9) requires the ground-truth conditional distribution $p(\boldsymbol{s}|\boldsymbol{c})$, whose closed form is unknown. Therefore, we use a probabilistic neural network $q_\theta(\boldsymbol{s}|\boldsymbol{c})$ to approximate $p(\boldsymbol{s}|\boldsymbol{c})$ by maximizing the log-likelihood $\mathcal{F}(\theta) = \sum_{u=1}^{M} \sum_{i=1}^{N_u} \log q_\theta(\boldsymbol{s}_{ui}|\boldsymbol{c}_{ui})$. With the learned $q_\theta(\boldsymbol{s}|\boldsymbol{c})$, the

objective for minimizing $\mathcal{I}(\boldsymbol{s}; \boldsymbol{c})$ becomes:

$$\hat{\mathcal{I}}_3 := \frac{1}{N} \sum_{u=1}^{M} \sum_{i=1}^{N_u} \Big[ \log q_\theta(\boldsymbol{s}_{ui}|\boldsymbol{c}_{ui}) - \frac{1}{N} \sum_{v=1}^{M} \sum_{j=1}^{N_v} \log q_\theta(\boldsymbol{s}_{ui}|\boldsymbol{c}_{vj}) \Big]. \tag{10}$$

When weights of encoders $E_c, E_s$ are updated, the embedding spaces $\boldsymbol{s}, \boldsymbol{c}$ change, which leads to the changing of conditional distribution $p(\boldsymbol{s}|\boldsymbol{c})$. Therefore, the neural approximation $q_\theta(\boldsymbol{s}|\boldsymbol{c})$ must be updated again. Consequently, during training, the encoders $E_c, E_s$ and the approximation $q_\theta(\boldsymbol{s}|\boldsymbol{c})$ are updated iteratively. In the Supplementary Material, we further discuss that with a good approximation $q_\theta(\boldsymbol{s}|\boldsymbol{c})$, $\hat{\mathcal{I}}_3$ remains an MI upper bound.

## 3.4 Encoder-Decoder Framework

With the aforementioned MI estimates $\hat{\mathcal{I}}_1$, $\hat{\mathcal{I}}_2$, and $\hat{\mathcal{I}}_3$, the final training loss of our method is

$$\mathcal{L}^* = [\hat{\mathcal{I}}_3 - \hat{\mathcal{I}}_1 - \hat{\mathcal{I}}_2] - \beta \mathcal{F}(\theta), \tag{11}$$

where $\beta$ is a positive number re-weighting the two objective terms. Term $\hat{\mathcal{I}}_3 - \hat{\mathcal{I}}_1 - \hat{\mathcal{I}}_2$ is minimized w.r.t the parameters in encoders $E_c, E_s$ and decoder $D$; term $\mathcal{F}(\theta)$ as the likelihood function of $q_\theta(\boldsymbol{s}|\boldsymbol{c})$ is maximized w.r.t the parameter $\theta$. In practice, the two terms are updated iteratively with gradient descent (by fixing one and updating another). The training and transfer processes of our model are shown in Figure 1. We name this MI-guided learning framework as **I**nformation-theoretic **D**isentangled **E**mbedding for **V**oice **C**onversion (IDE-VC).

## 4 Related Work

**Many-to-many Voice Conversion** Traditional voice style transfer methods mainly focus on one-to-one and many-to-one conversion tasks, which can only transfer voices into one target speaking style. This constraint limits the applicability of the methods. Recently, several many-to-many voice conversion methods have been proposed, to convert voices in broader application scenarios. StarGAN-VC (Kameoka et al., 2018) uses StarGAN (Choi et al., 2018) to enable many-to-many transfer, in which voices are fed into a unique generator conditioned on the target speaker identity. A discriminator is also used to evaluate generation quality and transfer accuracy. Blow (Serrà et al., 2019) is a flow-based generative model (Kingma & Dhariwal, 2018), that maps voices from different speakers into the same latent space via normalizing flow (Rezende & Mohamed, 2015). The conversion is accomplished by transforming the latent representation back to the observation space with the target speaker's identifier. Two other many-to-many conversion models, AUTOVC (Qian et al., 2019) and AdaIN-VC (Chou & Lee, 2019), extend applications into zero-shot scenarios, *i.e.*, conversion from/to a new speaker (unseen during training), based on only a few utterances. Both AUTOVC and AdaIN-VC construct an encoder-decoder framework, which extracts the style and content of one speech sample into separate latent embeddings. Then when a new voice from an unseen speaker comes, both its style and content embeddings can be extracted directly. However, as discussed in the Introduction, both methods do not have explicit regularizers to reduce the correlation between style and content embeddings, which limits their performance.

**Disentangled Representation Learning** Disentangled representation learning (DRL) aims to encode data points into separate independent embedding subspaces, where different subspaces represent different data attributes. DRL methods can be classified into unsupervised and supervised approaches. Under unsupervised setups, Burgess et al. (2018), Higgins et al. (2016) and Kim & Mnih (2018) use latent embeddings to reconstruct the original data while keeping each dimension of the embeddings independent with correlation regularizers. This has been challenged by Locatello et al. (2019), in that each part of the learned embeddings may not be mapped to a meaningful data attribute. In contrast, supervised DRL methods effectively learn meaningful disentangled embedding parts by adding different supervision to different embedding components. Between the two embedding parts, the correlation is still required to be reduced to prevent the revealing of information to each other. The correlation-reducing methods mainly focus on adversarial training between embedding parts (Hjelm et al., 2018; Kim & Mnih, 2018), and mutual information minimization (Chen et al., 2018; Cheng et al., 2020b). By applying operations such as switching and combining, one can use disentangled representations to improve empirical performance on downstream tasks, *e.g.* conditional generation (Burgess et al., 2018), domain adaptation (Gholami et al., 2020), and few-shot learning (Higgins et al., 2017).

## 5 EXPERIMENTS

We evaluate our IDE-VC on real-world voice a dataset under both many-to-many and zero-shot VST setups. The selected dataset is CSTR Voice Cloning Toolkit (VCTK) (Yamagishi et al., 2019), which includes 46 hours of audio from 109 speakers. Each speaker reads a different sets of utterances, and the training voices are provided in a non-parallel manner. The audios are downsampled at 16kHz. In the following, we first describe the evaluation metrics and the implementation details, and then analyze our model's performance relative to other baselines under many-to-many and zero-shot VST settings.

### 5.1 EVALUATION METRICS

**Objective Metrics** We consider two objective metrics: Speaker verification accuracy (*Verification*) and the Mel-Cepstral Distance (*Distance*) (Kubichek, 1993). The speaker verification accuracy measures whether the transferred voice belongs to the target speaker. For fair comparison, we used a third-party pre-trained speaker encoder Resemblyzer[1] to classify the speaker identity from the transferred voices. Specifically, style centroids for speakers are learned with ground-truth voice samples. For a transferred voice, we encode it via the pre-trained speaker encoder and find the speaker with the closest style centroid as the identity prediction. For the *Distance*, the vanilla Mel-Cepstral Distance (MCD) cannot handle the time alignment issue described in Section 2. To make reasonable comparisons between the generation and ground truth, we apply the Dynamic Time Warping (DTW) algorithm (Berndt & Clifford, 1994) to automatically align the time-evolving sequences before calculating MCD. This DTW-MCD distance measures the similarity of the transferred voice and the real voice from the target speaker. Since the calculation of DTW-MCD requires parallel data, we select voices with the same content from the VCTK dataset as testing pairs. Then we transfer one voice in the pair and calculate DTW-MCD with the other voice as reference.

**Subjective Metrics** Following Wester *et al.* (Wester et al., 2016), we use the naturalness of the speech (*Naturalness*), and the similarity of the transferred speech to target identity (*Similarity*) as subjective metrics. For *Naturalness*, annotators are asked to rate the score from 1-5 for each transferred speech. For the *Similarity*, the annotators are presented with two audios (the converted speech and the corresponding reference), and are asked to rate the score from 1 to 4. For both scores, the higher the better. Following the setting in Blow (Serrà et al., 2019), we report Similarity defined as a total percentage from the binary rating. The evaluation of both subjective metrics is conducted on Amazon Mechanical Turk (MTurk)[2]. More details about evaluation metrics are provided in the Supplementary Material.

### 5.2 IMPLEMENTATION DETAILS

Following AUTOVC (Qian et al., 2019), our model inputs are represented via mel-spectrogram. The number of mel-frequency bins is set as 80. When voices are generated, we adopt the WaveNet vocoder (Oord et al., 2016) pre-trained on the VCTK corpus to invert the spectrogram signal back to a waveform. The spectrogram is first upsampled with deconvolutional layers to match the sampling rate, and then a standard 40-layer WaveNet is applied to generate speech waveforms. Our model is implememted with Pytorch and takes 1 GPU day on an Nvidia Xp to train.

**Encoder Architecture** The speaker encoder consists of a 2-layer long short-term memory (LSTM) with cell size of 768, and a fully-connected layer with output dimension 256. The speaker encoder is initialized with weights from a pretrained GE2E (Wan et al., 2018) encoder. The input of the content encoder is the concatenation of the mel-spectrogram signal and the corresponding speaker embedding. The content encoder consists of three convolutional layers with 512 channels, and two layers of a bidirectional LSTM with cell dimension 32. Following the setup in AUTOVC (Qian et al., 2019), the forward and backward outputs of the bi-directional LSTM are downsampled by 16.

**Decoder Architecture** Following AUTOVC (Qian et al., 2019), the initial decoder consists of a three-layer convolutional neural network (CNN) with 512 channels, three LSTM layers with cell dimension 1024, and another convolutional layer to project the output of the LSTM to dimension of 80. To enhance the quality of the spectrogram, following AUTOVC (Qian et al., 2019), we use a post-network consisting of five convolutional layers with 512 channels for the first four layers, and

---

[1]https://github.com/resemble-ai/Resemblyzer
[2]https://www.mturk.com/

Table 1: Many-to-many VST evaluation results. For all metrics except Distance, higher is better.

| Metric | Objective | | Subjective | |
|---|---|---|---|---|
| | Distance | Verification[%] | Naturalness [1–5] | Similarity [%] |
| StarGAN | 6.73 | 71.1 | 2.77 | 51.5 |
| AdaIN-VC | 6.98 | 85.5 | 2.19 | 50.8 |
| AUTOVC | 6.73 | 89.9 | 3.25 | 55.0 |
| Blow | 8.08 | - | 2.11 | 10.8 |
| IDE-VC (Ours) | **6.70** | **92.2** | **3.26** | **68.5** |

Table 2: Zero-Shot VST evaluation results. For all metrics except Distance, higher is better.

| Metric | Objective | | Subjective | |
|---|---|---|---|---|
| | Distance | Verification[%] | Naturalness [1–5] | Similarity [%] |
| AdaIN-VC | 6.37 | 76.7 | 2.67 | 68.4 |
| AUTOVC | 6.68 | 60.0 | 2.19 | 58.6 |
| IDE-VC (Ours) | **6.31** | **81.1** | **3.33** | **76.4** |

80 channels for the last layer. The output of the post-network can be viewed as a residual signal. The final conversion signal is computed by directly adding the output of the initial decoder and the post-network. The reconstruction loss is applied to both the output of the initial decoder and the final conversion signal.

**Approximation Network Architecture** As described in Section 3.3, minimizing the mutual information between style and content embeddings requires an auxiliary variational approximation $q_\theta(s|c)$. For implementation, we parameterize the variational distribution in the Gaussian distribution family $q_\theta(s|c) = \mathcal{N}(\mu_\theta(c), \sigma_\theta^2(c) \cdot \mathbf{I})$, where mean $\mu_\theta(\cdot)$ and variance $\sigma_\theta^2(\cdot)$ are two-layer fully-connected networks with $\tanh(\cdot)$ as the activation function. With the Gaussian parameterization, the likelihoods in objective $\hat{\mathcal{I}}_3$ can be calculated in closed form.

## 5.3 STYLE TRANSFER PERFORMANCE

For the many-to-many VST task, we randomly select 10% of the sentences for validation and 10% of the sentences for testing from the VCTK dataset, following the setting in Blow (Serrà et al., 2019). The rest of the data are used for training in a non-parallel scheme. For evaluation, we select voice pairs from the testing set, in which each pair of voices have the same content but come from different speakers. In each testing pair, we conduct transfer from one voice to the other voice's speaking style, and then we compare the transferred voice and the other voice as evaluating the model performance. We test our model with four competitive baselines: Blow (Serrà et al., 2019)[3], AUTOVC (Qian et al., 2019), AdaIN-VC (Chou & Lee, 2019) and StarGAN-VC (Kameoka et al., 2018). The detailed implementation of these four methods are provided in the Supplementary Material. Table 1 shows the subjective and objective evaluation for the many-to-many VST task. Both methods with the encoder-decoder framework, AdaIN-VC and AUTOVC, have competitive results. However, our IDE-VC outperforms the other baselines on all metrics, demonstrating that the style-content disentanglement in the latent space improves the performance of the encoder-decoder framework.

For the zero-shot VST task, we use the same train-validation dataset split as in the many-to-many setup. The testing data are selected to guarantee that no test speaker has any utterance in the training set. We compare our model with the only two baselines, AUTOVC (Qian et al., 2019) and AdaIN-VC (Chou & Lee, 2019), that are able to handle voice transfer for newly-coming unseen speakers. We used the same implementations of AUTOVC and AdaIN-VC as in the many-to-many VST. The evaluation results of zero-shot VST are shown in Table 2, among the two baselines AdaIN-VC performs better than AUTOVC overall.Our IDE-VC outperforms both baseline methods, on all metrics. All three tested models have encoder-decoder transfer frameworks, the superior performance

---

[3]For Blow model, we use the official implementation available on Github (https://github.com/joansj/blow). We report the best result we can obtain here, under training for 100 epochs (11.75 GPU days on Nvidia V100).

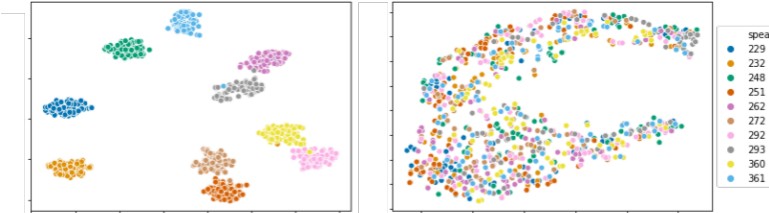

Figure 2: Left: t-SNE visualization for speaker embeddings. Right: t-SNE visualization for content embedding. The embeddings are extracted from the voice samples of 10 different speakers.

of IDE-VC indicates the effectiveness of our disentangled representation learning scheme. More evaluation details are provided in the supplementary material.

## 5.4 DISENTANGLEMENT DISCUSSION

Besides the performance comparison with other VST baselines, we demonstrate the capability of our information-theoretic disentangled representation learning scheme. First, we conduct a t-SNE (Maaten & Hinton, 2008) visualization of the latent spaces of the IDE-VC model. As shown in the left of Figure 2, style embeddings from the same speaker are well clustered, and style embeddings from different speakers separate in a clean manner. The clear pattern indicates our style encoder $E_s$ can verify the speakers' identity from the voice samples. In contrast, the content embeddings (in the right of Figure 2) are indistinguishable for different speakers, which means our content encoder $E_c$ successfully eliminates speaker-related information and extracts rich semantic content from the data.

We also empirically evaluate the disentanglement, by predicting the speakers' identity based on only the content embeddings. A two-layer fully-connected network is trained on the testing set with a content embedding as input, and the corresponding speaker identity as output. We compare our IDE-VC with AUTOVC and AdaIN-VC, which also output content embeddings. The classification results are shown in Table 3. Our IDE-VC reaches the lowest classification accuracy, indicating that the content embeddings learned by IDE-VC contains the least speaker-related information. Therefore, our IDE-VC learns disentangled representations with high quality compared with other baselines.

Table 3: Speaker identity prediction accuracy on content embedding.

|  | Accuracy[%] |
| --- | --- |
| AUTOVC | 9.5 |
| AdaIN-VC | 19.0 |
| IDE-VC | **8.1** |

## 5.5 ABLATION STUDY

Moreover, we have considered an ablation study that addresses performance effects from different learning losses in (11), with results shown in Table 4. We compare our model with two models trained by part of the loss function in (11), while keeping the other training setups unchanged, including the model structure. From the results, when the model is trained without the style encoder loss term $\hat{\mathcal{I}}_1$, a transferred voice still is generated, but with a large distance to the ground truth. The verification accuracy also significantly decreases with no speaker-related information utilized. When the disentangling term $\hat{\mathcal{I}}_3$ is removed, the model still reaches competitive performance, because the style encoder $E_s$ and decoder $D$ are well trained by $\hat{\mathcal{I}}_1$ and $\hat{\mathcal{I}}_2$. However, when adding term $\hat{\mathcal{I}}_3$, we disentangle the style and content spaces, and improve the transfer quality with higher verification accuracy and less distortion. The performance without term $\hat{\mathcal{I}}_2$ is not reported, because the model cannot even generate fluent speech without the reconstruction loss.

Table 4: Ablation study with different training losses. Performance is measured by objective metrics.

|  | Distance | Verification[%] |
| --- | --- | --- |
| Without $\hat{\mathcal{I}}_1$ | 9.81 | 11.1 |
| Without $\hat{\mathcal{I}}_3$ | 6.73 | 89.4 |
| IDE-VC | **5.66** | **92.2** |

## 6 CONCLUSIONS

We have improved the encoder-decoder voice style transfer framework by disentangled latent representation learning. To effectively induce the style and content information of speech into independent embedding latent spaces, we minimize a sample-based mutual information upper bound between style and content embeddings. The disentanglement of the two embedding spaces ensures the voice transfer accuracy without information revealed from each other. We have also derived two new multi-group mutual information lower bounds, which are maximized during training to enhance the representativeness of the latent embeddings. On the real-world VCTK dataset, our model outperforms previous works under both many-to-many and zero-shot voice style transfer setups. Our model can be naturally extended to other style transfer tasks modeling time-evolving sequences, *e.g.*, video and music style transfer. Moreover, our general multi-group mutual information lower bounds have broader potential applications in other representation learning tasks.

## ACKNOWLEDGEMENTS

This research was supported in part by the DOE, NSF and ONR.

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
