# OpenReview forum: "Improving Zero-Shot Voice Style Transfer via Disentangled Representation Learning"
_ICLR.cc/2021/Conference — ICLR 2021 Poster_

### Official Review · AnonReviewer2 · 2020-10-25
**Review by AnonReviewer2**

**Rating:** 6
**Confidence:** 5

**Review:**

This submission proposes a training approach for voice style transfer using encoder-decoder framework and content and style representations. The approach combines multiple mutual-information (MI) based terms into a single objective function. One of the MI based terms is the MI between content and style representations. By minimising mutual information between these representations, the training approach yields models where these representations are disentangled. Experimental results show that this approach leads to improved performance in speaker verification and speech similarity tasks. Experimental results in challenging zero-shot conditions also demonstrate improved performance in speaker verification, speech naturalness and speech similarity tasks.

Quality: The quality of this submission appears to be generally OK. There are few cases I would like to draw your attention to:

1) Although I understand the desire to give a theoretic flavour to this work, I am quite concerned that you boiled down all of the information theory into the mutual information.
2) "our ... outperforms the other baselines on all metrics" is a very problematic statement to make. First of all, it is a very uninformative statement, I can see numbers myself. Second, it is a very poor analysis - I learnt nothing from this as you provide no more information. Third, some gains are marginal so such claim is not very strong. Fourth, I am more interested if what you expect to be better at compared to AUTOVC actually transpires into gains.
3) Based on Table 3 you make a claim that your approach disentangles representations with higher quality compared to other baselines. I am sorry but I do not see that and am surprised why would you say that about accuracies of 8.1 and 9.5%?
4) I am also surprised not to see discussion about work done on learning language, speaker, etc invariant representations in speech recognition.

Clarity: The clarity of this submission appears to be generally OK. There are few cases I would like to draw your attention to:

1) Given statements such as "good approximation", you must make it absolutely unambiguous whether the final loss you are optimising is a bound or not;
2) You are throwing "based on the MI data-processing inequality, we conclude" in a passing by manner on page 3. I do not consider it to be an appropriate line of argument or explanation. You need to discuss this aspect in a significantly more details.
3) It is very unclear how much credit to give you for deriving these bounds given that they are based on other bounds.

Originality: I think this work is moderately original.

Significance: I think this work will have a moderate significance.

Pros:

1) Using MI to learn disentangled representations is an interesting idea that may give rise to follow up works.
2) Zero-shot experiments show that this approach clearly outperforms baselines based on 3 out of 4 metrics.

Cons:

1) This submission does not really make a convincing case for using MI to learn disentangled representations generally.
2) It also does not really strike a good balance in terms of theory versus engineering. The engineering part, which is way more important for this submission (at no point you go back to the theory in your submission), is left with a very limited space and very limited discussion about options available.

---

> ### Author Response · Authors · 2020-11-25
> **Response to Reviewer2**
>
> Thank you for your constructive feedback. As for your concerns:
>
> 1. About **Theoretic Part**: Our initial point is to utilize the disentangled representation learning to enhance the voice style transfer tasks. While building our framework, we found a connection between our encoder-decoder model and mutual information optimization. This is also part of our contribution that we provided an information-theoretic justification to the proposed model.
>
> 2. About **Empirical Gains**: In our framework, we disentangled the style and content embeddings into (ideally) independent spaces, so that when conducting style transfer, our learned model can ensure that no extra information is revealed (i.e. No style information included in content embeddings and vice versa). As mentioned in the introduction, AUTOVC has no regularizer to guarantee that the content encoder does not encode any style information. The empirical gains of IDE-VC compared to AUTOVC demonstrate the effectiveness of our disentangling framework.
>
> 3. About **Disentangling Quality**: The classification result between our method and AUTOVC might seem not significant to the reviewer because we report the classification result among 20 speakers. The 20 classification categories scaled down the difference of models’ accuracy (note that a completely random guess has 5% accuracy). To further show the performance difference, we conducted additional experiments about speaker-id classification among 3 speakers with 80 voice samples for each speaker (note that a completely random guess has 33.3% accuracy). In this three-class classification, our model achieved 43% while AUTOVC is 70%, which supports our claim that the disentangling framework better eliminates style information from content embeddings.
>
> 4. About **Relative Work**: We will add more related work on learning language, speaker, and invariant representation in the revision.
>
> 5. About **MI Approximation**: Term I1 and I2 are strictly mutual information lower bounds. Term I3 can either remain a mutual information upper bound or provide a good approximation with absolute error bounded by the KL-divergence between the ground-truth distribution p(s|c) and variational distribution q(s|c). By maximizing the log-likelihood of q(s|c), we can minimize the KL-divergence between p(s|c) and q(s|c), so that I3 can become a reliable MI estimator.
>
> 6. About **Data-processing Inequality**: The data-processing inequality refers to a theorem in information theory, whose detailed statement can be found at https://en.wikipedia.org/wiki/Data_processing_inequality. The condition of the inequality is satisfied for u->x->s, because of p(s|x, u)=p(s|x) with x including sufficient information to infer s. We have also added a statement to the inequality in the supplementary material.
>
> 7. About **Derived Bounds**: The bounds are derived based on prior works. Our contribution is to use the derived bounds to provide a theoretical justification to the encoder-decoder style transfer frameworks.

---

### Official Review · AnonReviewer3 · 2020-10-28
**Previous concerns not fully addressed**

**Rating:** 6
**Confidence:** 5

**Review:**

This paper proposes a zero-shot voice style transfer (VST) algorithms that explicitly controls the disentanglement between content information and style information. Experiments show that the proposed algorithm can achieve significant improvement over the existing state-of-the-art VST algorithms. There are two major strengths of this paper. First, it motivates the algorithm design from an information-theoretic perspective. Second, the performance improvement is significant.

However, since it is a resubmission from a previous machine learning conference, the previous concerns regarding limited novelty are not fully addressed. More specifically, if we view the proposed algorithm entirely from a technical perspective, there are two major innovations over AutoVC: 1) The style embedding is trained together with the content embedding, instead of being pre-trained; 2) The introduction of I3. The latter does not seem fully justified.

First, it is shown in Table 4 that without I3, the drop in performance is not obvious. The authors ascribe this to that I1 and I2 already suffice to train the good model. Does it mean that the introduction of I3 is not as important an innovation as co-training?

Second and more importantly, in all the experiments, the proposed system retains AutoVC’s physical bottleneck design, which was the key to disentangling style in AutoVC. In order to justify that I3 is a better disentangling mechanism than AutoVC, it is necessary to perform an ablation study where the bottleneck is widened and see if I3 still guarantees disentanglement, without which it is hard to justify the value of I3.

Besides the concern regarding novelty, there are a few other concerns. There lacks a back-to-back comparison in Figure 2. What do the embeddings look like for AdaINVC and AutoVC? Also, Figure 2 only shows that the content embedding does not include style information. It would also be helpful to show the style embedding does not include content information by showing the content embedding and style embedding cluster with respect to different phones.

To sum, without strong supporting evidence for the novel design in IDE-VC, it is hard to judge the contribution of this paper. I would look forward to more thorough evaluations in the rebuttal.

---

> ### Author Response · Authors · 2020-11-25
> **Response to Reviewer3**
>
> Thank you for your thoughtful feedback. As for your concerns:
>
> About **Ablation Study** in Table 4: We conducted one more ablation study under the zero-shot VST setup, in which Distance and Verification of model without I1 are 9.49 and 10.3; model without I3 are 6.71 and 62.5; the whole model (I1+I2+I3) are 6.31 and 81.1.
> From the results, the performance margin is more significant than many-to-many setups, which further supported that I3, as a disentangling term, enhanced the voice conversion.
>
> About **Bottleneck Design**: We kept the same bottleneck design as AUTOVC to have a fair comparison of the effectiveness of our disentangled learning scheme. To alleviate the reviewer’s concern, we further conducted an ablation study in which the bottleneck is widened. Specifically, we use a sampling rate = 4 and conduct experiments under the zero-shot setup. Our proposed model achieved the Distance of 6.24 and Verification as 80. While for AUTOVC, the distance is 6.59 and verification is 41. The results demonstrated that the bottleneck design has little impact on the disentanglement ability of the proposed model.
>
> About **T-SNE Plot**: We have updated visualization of content embeddings from our model and AUTOVC in the supplementary material as in Figure 3 and Figure 4. The embeddings are extracted from 3 speakers with a sampling rate = 4. From the plot, we found the T-SNE of AUTOVC has a pattern related to the speaker-id, which means the speaker information is not well-eliminated. The T-SNE plot of style embedding in Figure 2 is sufficient to conclude that the style embedding does not include content information. Because if the style embedding contains content information, in the T-SNE, the style embedding would not be well-clustered.
>
> We will update the new experimental results to the main draft in later revisions.

---

### Official Review · AnonReviewer4 · 2020-11-03

**Rating:** 6
**Confidence:** 1

**Review:**

The paper proposes a way to do voice conversion by learning a disentangled representation of the style and content of the audio. This disentangled representation is enforced by minimizing the mutual information between these the style and content encodings of the audio. The paper shows maths to derive its loss functions.

Pros:
1. The paper looks novel.
2. The results section shows improvement over the chosen baseline.

Cons:
1. Very difficult to understand/follow the central part of the paper.
2. Associated samples would have been great to bolster the claims.

How does this work relate to other representation learning methods?

I think the paper in the current form might be interesting to the ICLR community.

---

> ### Author Response · Authors · 2020-11-25
> **Response to Reviewer4**
>
> Thank you for your important suggestions and comments. As for your comments:
>
> About **Central Part of the Paper**: We provide an intuitive description of our framework in Section 3.1, where we aim to encode input voices into disentangled latent representations as style embeddings and content embeddings. Then Section 3.2 and 3.3 describe the detailed derivation of our objectives. We made some revisions to clarify the structure of this section.
>
> About **Voice Samples**: We have provided the link to transferred samples in Section B of the supplementary material. Please check the voice samples at https://idevc.github.io/.
>
> About **Relative Work**: We provide the relative work of our disentangled framework in section 4. There are several prior works using mutual information to enhance disentangled representation learning. Our work is the first model to apply disentangled representation learning into voice style transfer scenarios.

---

### Official Review · AnonReviewer5 · 2020-11-07
**Interesting, mostly well-written paper, that I found slightly hard to follow.**

**Rating:** 7
**Confidence:** 2

**Review:**

The paper presents a framework for "disentangling" style and content from audio samples, a very interesting topic. It's a good read until the math becomes a bit too hard/convoluted for me to follow -- it would help me, for one, if more intuitions were given and if the bottom line were explained more succinctly. Intuitions are actually given along the way, but I found the overall approach slightly beyond me -- or at least requiring considerable effort to piece together. My impression is that the approach is coherent, though, and the results are good. The citation of previous work is good.

A number of specific comments follow.

Abstract: "On real-world datasets, our method outperforms other baselines and obtains state-of-the-art results in terms of
transfer accuracy and voice naturalness."

Intro: "Experiments demonstrate that our method outperforms previous works under both many-to-many and zero-shot transfer setups."

Be more specific, what kinds of experiments, what datasets, what metrics?

"Among the a few existing models that address ..." --> remove superfluous "a"

"To transfer a source xui from speaker u to the target style of the voice of speaker v, xvj ...": --> " ... to the target style of the voice of speaker v as reflected in a particular utterance j, from speaker v, ..."? I could imagine the style of speaker v to be considered in aggregate, averaging somehow over all utterances from that speaker.

"(i.e., ideally, s contains rich style information of x with no content information, and vice versa)" --> "(i.e., ideally, s contains rich style
information of x with no content information, and vice versa for c)"

"... is the mean of all style embedding... " --> embeddingS

"Under unsupervised setups, Burgess et al. (2018); Higgins et al. (2016); Kim & Mnih (2018) use... " --> separate those references with commas, not semi-colons, i.e. --> "... A, B and C use ..."

Eq. (7), give intuition for the e^{-1} term?

"Based on the criterion for s in equation (7), a well-learned style encoder Es pulls all style embeddings sui from speaker u together": I am not really following here, it starts out well with the intro to Theorem 3.1, but then I'm confused as to what is an upper/lower bound for what and that the actual criterion is. I don't quite know specifically what the authors mean by, "a well-learned style encoder Es pulls all style embeddings sui from speaker u together". I realize this work summaries derivations in the supplementary material, but ideally the central intuitions and actual specific bottom-line criteria used would be much clearer.

Fig 2b: show on the figure the use of I2 and I3, separate from I? Or at least, in the text, clarify how I2 is used? (I3 is mentioned).

---

> ### Author Response · Authors · 2020-11-25
> **Response to Reviewer5**
>
> Thank you for your supportive comments. As for your concerns:
>
> About **More Specific Description**: We conduct voice style transfer experiments under both many-to-many and zero-shot setups on the VCTK dataset, and test the transfer accuracy and voice naturalness compared with other baselines. Also, we have updated the description in the abstract and introduction.
>
> About **Typos**: Thanks for the detailed comments. We have updated them in the revision.
>
> About **Equation (7)**: The objective in equation (7) is derived based on the NWJ mutual information bound, which is described in Eq. (2). Therefore, the term e^{-1} in Eq.(7) is inherited from Eq. (2).
>
> About **Theorem 1**: The objective in Theorem 1 is a lower bound to the mutual information between style embedding and input voice. As mentioned in Section 3.1, the exact mutual information is hard to calculate only with data samples, therefore we maximize a lower bound to maximize the mutual information between style embedding and input voice.
>
> About **Figure 2b**: We have updated the description to I3 in the revision.

---

### Decision · Program_Chairs · 2021-01-07
**Final Decision**

**Decision:**

Accept (Poster)

**Comment:**

The authors propose a new model to learn voice style transfer using an encoder-decoder framework with the aim of disentangling content and style representations.

The strengths of the paper are:
+ the method is well-motivated with sound theoretical justification
+ the authors improve up on the prior work by augmenting the loss with an information-theoretic term
+ empirical evaluations demonstrate performance improvements in speaker verification and speech similarity tasks
+ demonstrate improvements in the challenging zero-shot task

Several reviewers requested improvements in readability
+ “ideally the central intuitions and actual specific bottom-line criteria used would be much clearer.”
+ more clarify on empirical details including challenges that needed to be addressed